# Sec20 Is Required for Autophagic and Endocytic Degradation Independent of Golgi-ER Retrograde Transport

**DOI:** 10.3390/cells8080768

**Published:** 2019-07-24

**Authors:** Zsolt Lakatos, Péter Lőrincz, Zoltán Szabó, Péter Benkő, Lili Anna Kenéz, Tamás Csizmadia, Gábor Juhász

**Affiliations:** 1Department of Anatomy, Cell and Developmental Biology, Eötvös Loránd University, H-1117 Budapest, Hungary; 2Premium Postdoctoral Research Program, Hungarian Academy of Sciences, H-1117 Budapest, Hungary; 3Institute of Genetics, Biological Research Centre, Hungarian Academy of Sciences, H-6726 Szeged, Hungary

**Keywords:** autophagy, *Drosophila*, endocytosis, lysosome, secretion, Sec20, Syx18

## Abstract

Endocytosis and autophagy are evolutionarily conserved degradative processes in all eukaryotes. Both pathways converge to the lysosome where cargo is degraded. Improper lysosomal degradation is observed in many human pathologies, so its regulatory mechanisms are important to understand. Sec20/BNIP1 (BCL2/adenovirus E1B 19 kDa protein-interacting protein 1) is a BH3 (Bcl-2 homology 3) domain-containing SNARE (soluble N-ethylmaleimide-sensitive factor-attachment protein receptors) protein that has been suggested to promote Golgi-ER retrograde transport, mitochondrial fission, apoptosis and mitophagy in yeast and vertebrates. Here, we show that loss of Sec20 in *Drosophila* fat cells causes the accumulation of autophagic vesicles and prevents proper lysosomal acidification and degradation during bulk, starvation-induced autophagy. Furthermore, Sec20 knockdown leads to the enlargement of late endosomes and accumulation of defective endolysosomes in larval *Drosophila* nephrocytes. Importantly, the loss of Syx18 (Syntaxin 18), one of the known partners of Sec20, led to similar changes in nephrocytes and fat cells. Interestingly. Sec20 appears to function independent of its role in Golgi-ER retrograde transport in regulating lysosomal degradation, as the loss of its other partner SNAREs Use1 (Unconventional SNARE In The ER 1) and Sec22 or tethering factor Zw10 (Zeste white 10), which function together in the Golgi-ER pathway, does not cause defects in autophagy or endocytosis. Thus, our data identify a potential new transport route specific to lysosome biogenesis and function.

## 1. Introduction

Lysosomal degradation is crucial for normal homeostasis and survival of all eukaryotic cells. Lysosomes receive cargo mainly from autophagy and endocytosis [1,2]. Defective lysosomes are often observed in several human pathologies, such as Alzheimer’s disease [3]; thus, understanding how lysosomal function is regulated is of high clinical importance. A key step of the main pathway of autophagy and endocytosis is the fusion events happening between autophagosomes or late endosomes and lysosomes to form degradative auto- and endolysosomes, respectively [4,5]. Central mediators of vesicular fusion events are the SNARE (soluble N-ethylmaleimide-sensitive factor-attachment protein receptors) proteins and the tethering factors that dock vesicles into an adjacent state and promote fusion [6].

SNARE proteins are classified into four groups based on the amino acid found in the zero layer of the assembled trans-SNARE complex; thus, there are R-SNAREs and Q_a,b,c_-SNAREs [7]. Sec20 or its mammalian orthologue BNIP1 (BCL2/adenovirus E1B 19 kDa protein-interacting protein 1 [8]) is a Q_b_-SNARE that was first described in yeast as a factor mediating the fusion of Golgi vesicles with the ER in a retrograde transport route [9]. Its partners include Sec22, Syx18 (Syntaxin 18) and Use1 (Unconventional SNARE In The ER 1) as part of the SNARE complex, and Dsl1 (Dependence on SLy1-20) as a tethering protein [10,11,12]. Sec20 has been implicated in other processes as well such as apoptosis as it contains a BH3 (Bcl-2 homology 3) domain, which is a feature of apoptosis-regulating Bcl2-like (B-cell lymphoma 2-like) proteins [13,14,15,16,17]. Indeed, the overexpression of BNIP1 increases the number of apoptotic cells in the retina of zebrafish [18]. Sec20 may also have a role in mitophagy as it localizes partly to mitochondria and its overexpression results in the fragmentation of the mitochondrial network [19,20]. Although the potential role of Sec20 is not known in starvation-induced autophagy, a study showed that the level of *bnip1* mRNA is increased in rat neocortical neurons upon starvation, which is also seen for many autophagy-related genes [21].

Here, we characterize Sec20 as a new regulator of autophagic and endocytic degradation in *Drosophila*, and propose that it acts independent of its role in Golgi-ER retrograde transport based on a series of functional analyses.

## 2. Materials and Methods

### 2.1. Fly Stocks

Flies were raised at 25 °C on regular corn meal-agar-yeast medium. Sec20 RNAi1 (FlyBase ID: FBst0034693), Use1 RNAi (FlyBase ID: FBst0043253), Sec22 RNAi (FlyBase ID: FBst0034893), Syx18 RNAi (FlyBase ID: FBst0026721) and Zw10 RNAi (FlyBase ID: FBst0042643), GMR-Gal4 (FlyBase ID: FBst0001104) were obtained from Bloomington Drosophila Stock Center, Bloomington, IN, USA. Sec20 RNAi2 (2023R-1) was obtained from Fly Stocks of National Institute of Genetics (Nig-Fly), Mishima, Japan. Sec20 RNAi3 (FlyBase ID: FBst0472138) was obtained from Vienna Drosophila Resource Center, Vienna, Austria. We generated Gal4-expressing fat cell clones using the following stocks: hs-Flp [22]; UAS-DCR2, Act > CD2 > Gal4; UAS-GFP, or hs-Flp [22]; dLamp-3x-mCherry, UAS-GFP; Act > CD2 > Gal4, UAS-DCR2, or hs-Flp [22]; 3x-mCherry-Atg8a, UAS-GFP; Act > CD2 > Gal4, UAS-DCR2 [23], and hs-Flp [22]; UAS-GFP-Lamp1; Act > CD2 > Gal4, UAS-DCR2, the UAS-GFP-Lamp1 line was kindly provided by Helmut Krämer (UT Southwestern Medical Center, Dallas, TX, USA) [24,25]. UAS-Vps41-9xHA was generated in our lab [26]. We estimated autophagic flux by the tandem GFP-mCherry-Atg8a reporter as described earlier [27,28]. Gal4 driver prospero-Gal4 for expression in nephrocytes was a gift of Bruce Edgar (ZMBH, Heidelberg, Germany). Fat cell-specific Gal4 driver was a gift from Thomas Neufeld (UMN, Minneapolis, MN, USA). Ey-Gal4 was a gift of Viktor Billes (Department of Genetics, Eötvös Loránd University, Budapest, Hungary). For our salivary gland experiments, we used the sgs3-Glue-GFP [29], sgs3-Glue-dsRed; fkh-Gal4 line [30]. The sgs3-Glue-dsRed reporter was provided by Andrew Andres (UNLV, NV, USA) [31]. The fkh-Gal4 was provided by Eric Baehrecke (UMMS, Worcester, MA, USA).

### 2.2. Larval Stages

In order to acquire starving fat cells, 95-h-old larvae were starved in 20% sucrose solution for 4 h at RT. For obtaining wandering fat tissue where developmental autophagy occurs, and for the nephrocyte experiments, wandering L3 instar larvae were dissected.

### 2.3. Immunohistochemistry

For immunostaining, larvae were dissected in phosphate buffered saline (PBS) and fixed with 4% formaldehyde in PBS (50 min at room temperature (RT)). Samples were extensively washed (3 × 10 min at RT) and permeabilized in PBS with 0.1% Triton X-100 (PBTX) for 10 min at RT, followed by incubation in blocking solution (5.0% FCS in PBTX, 30 min at RT). Samples were incubated in primary antibodies diluted in the blocking solution overnight at 4 °C. Samples were rinsed 3 times and washed with PBTX with extra 4% NaCl for 15 min at RT, then washed with PBTX for 3 × 15 min in PBTX at RT, then incubated in blocking solution for 30 min at RT, followed by incubation with secondary antibodies in blocking solution for 3 h at RT. Washing steps were repeated, nuclei were counterstained with DAPI (4′,6-diamidino-2-phenylindole) added to the NaCl-containing washing step, and samples were mounted in VECTASHIELD (Vector Laboratories, Burlingame, CA, USA, H-1000). The antibodies used were: monoclonal mouse anti-Rab7 (1:10, Developmental Studies Hybridoma Bank/DSHB, Iowa City, IA, USA,) [32], polyclonal rabbit anti-CathL (1:100; ab58991, Abcam, Cambridge, UK), rabbit anti-HA 1:100 (Merck, Darmstadt, Germany, H6908), polyclonal rabbit anti-Rbsn-5 (1:1000) [33], polyclonal rat anti-Atg8a (1:300) [34], polyclonal rabbit anti-p62/Ref(2)p (1:1000) [35], polyclonal goat anti-GMAP (1:1000, DSHB) [32],. Secondary antibodies were: Alexa Fluor 568 donkey anti-goat, Alexa Fluor 568 goat anti­rabbit, Alexa Fluor 568 goat anti-rat, Alexa Fluor 568 goat anti-mouse, Alexa Fluor 488 goat anti-rabbit, Alexa Fluor 488 goat anti-rat, Alexa Fluor 488 donkey anti-rabbit (all 1:1000; Invitrogen, Carlsbad, CA, USA).

### 2.4. LysoTracker Staining, Uptake Assays and Salivary Gland Dissection

For TexasRed-Avidin and DQ Red BSA uptake assays, L3 larval proventriculi with garland nephrocytes were prepared in cold Shields and Sang M3 insect medium (Merck, Darmstadt, Germany, S8398) and incubated in 0.1 mg/mL TexasRed-Avidin D (Vector Laboratories, A-2006) or 10 µg/mL DQ Red BSA (Thermo Fisher Scientific, Waltham, MA, USA, D12051) containing M3 for 5 min at RT, rinsed 3 times, and incubated in M3 for 30 min where stated, then fixed with 4% formaldehyde in PBS (50 min at RT). Samples were washed 3 × 10 min in PBS, stained with DAPI and mounted as described before. For LysoTracker staining, fat bodies were dissected in cold PBS and incubated in LysoTracker Red DND-99 (1:1000 in PBS; Thermo Fisher Scientific, L7528) for 1 min at RT. Samples were rinsed 3 times, mounted in 80% glycerol in PBS containing DAPI, and photographed immediately. Salivary glands were dissected in cold PBS, then fixed with 4% formaldehyde in PBS (5 min at RT), and mounted in 80% glycerol in PBS containing DAPI.

### 2.5. Fluorescent Imaging

Fluorescent images were obtained at RT with an AxioImager.M2 microscope (Carl Zeiss, Oberkochen, Germany) with an ApoTome2 grid confocal unit (Carl Zeiss) using EC Plan-Neofluar 40×/0.75-NA Air (Carl Zeiss) or Plan-Apochromat 40×/0.95-NA Air (Carl Zeiss) objectives for fat cells, and Plan-Apochromat 63×/1.40-NA Oil (Carl Zeiss) objective for nephrocytes, an Orca Flash 4.0 LT sCMOS camera (Hamamatsu Photonics, Hamamatsu, Japan), and ZEISS Efficient Navigation 2 software (Carl Zeiss). Immersol 518F (Carl Zeiss) immersion oil was used for the 63× objective. Images from 8 consecutive focal planes (section thickness: 0.25 µm for nephrocytes and 0.35 µm for fat cells) were projected onto one single image, except for the colocalization assays, where we aimed to exclude any false positive colocalization. Salivary gland images were taken using an AxioImager Z1 microscope (Carl Zeiss) at RT equipped with an Apotome1 grid confocal unit and an AxioCam MRm camera and an EC Plan-Neofluar 40×/0.75-NA objective and AxioVision SE64 Rel. 4.9.1 (Carl Zeiss) software. Images were processed in ZEISS Efficient Navigation 2 (Carl Zeiss) and Photoshop CS4 or CS6 (Adobe, San Jose, CA, USA) to present final figures. Compound eye pictures were taken using a Lumar V12 stereomicroscope (Carl Zeiss) equipped with AxioCam ERc5s camera (Carl Zeiss).

### 2.6. Electron Microscopic Analysis

Dissected fat bodies or nephrocytes were fixed in immunofixating solution containing 3.2% Paraformaldehyde, 0.5% or 1% glutaraldehyde for nephrocytes or fat cells, respectively, 1% sucrose, and 0.028% CaCl2 in 0.1 N sodium cacodylate, pH 7.4, overnight at 4 °C. For the correlative light and electronmicroscopic analysis in the mosaic EM assay, Sec20 RNAi, GFP-positive cells were detected as the GFP channel was photographed immediately after the start of the fixation using an AxioImager Z1 microscope (Carl Zeiss), AxioCam MRm camera and EC Plan-Neofluar 10×/0.3-NA and AxioVision SE64 Rel. 4.9.1 (Carl Zeiss) software Samples were postfixed and contrasted in 0.5% OsO_4_ for 1 h and in half-saturated aqueous uranyl acetate for 30 min at RT, dehydrated in a graded series (50%, 70%, 96% and 100%) of ethanol, and embedded in Durcupan (Fluka, Buchs, Switzerland) according to the manufacturer’s protocols. Then, 70-nm sections were stained in Reynolds lead citrate and viewed on a JEM-1011 transmission electron microscope (Jeol, Tokyo, Japan) equipped with a Morada digital camera (Olympus, Tokyo, Japan) using iTEM software (Olympus). Images were processed in Photoshop CS4 (Adobe) to present final figures.

### 2.7. Statistics

We quantified our unmodified, single focal plane images using ImageJ software (NIH, Bethesda, MD, USA); 10 cells of each genotype were quantified. For measurement, cells were randomly selected, and the areas or distances were measured manually. Thresholds were set by the same person in all images of a given type of experiment. In the mosaic animal experiments, a randomly generated RNAi cell clone and a neighboring control cell was chosen for quantification. The normality test for each data sets was performed using SPSS17 (IBM, Armonk, NY, USA), and as the distribution of at least one of the compared datasets was non-Gaussian in each analysis, we either performed the Mann-Whitney U-test for two samples or the Kruskal-Wallis test with post-hoc pairwise analysis for more than two samples.

## 3. Results

### 3.1. Sec20 Depletion Causes Defects during Endocytosis and Autophagy

The liver- and fat-like larval *Drosophila* fat tissue is commonly used to assess starvation-induced and developmental autophagy, and its larval garland nephrocytes are similar to human podocytes and proximal tubule cells and are excellent for the study of endocytosis [36,37,38,39].

Silencing of Sec20 (also known as CG2023 in *Drosophila*) increased the size of Rab7-positive late endosomes in larval nephrocytes (and the cells were bigger as well), while the size and distribution of Rbsn-5-positive early endosomes remained similar to controls (Figure 1A,B,G and Appendix A).

These phenotypes point to the impairment of endolysosomal degradation as nephrocytes carrying mutations affecting endosomal traffic often swell due to the continuous uptake of extracellular fluid which they fail to degrade [26,39]. The enlargement of late endosomes in Sec20 RNAi cells is a very similar phenotype to those mutants that lack a critical factor of late endosomal-lysosomal fusion (such as HOPS (Homotypic fusion and vacuole protein sorting) or Rab2 loss-of-function) [25,39,40,41]. We verified our results using two other independent Sec20 RNAi constructs that showed the same defect as the one we used throughout this work (Appendix A, and Figure 1G).

This phenotype most likely results from the continuous input to late endosomes in the absence of proper lysosomal fusion or degradation [39]. To address this, we incubated live nephrocytes in a solution containing a fluorescent tracker (TexasRed-Avidin) for 5 min, which is taken up and transported to the inner layers of the cells in normal circumstances (Figure 1C,H). However, the tracer gets trapped in the periphery of Sec20-depleted cells (Figure 1D,H). To see if the tracer would eventually be transported to the inner layers of the cells, in another experiment we chased the tracer for 30 min after the 5 min pulse. We found that although the tracer was transported closer to the perinuclear region of the nephrocytes even upon Sec20 depletion, it accumulated in larger vesicles than in the control cells (Figure 1E,F,H), suggesting that it accumulates in the enlarged late endosomes.

In order to assess whether there are functional endolysosomes in the nephrocytes, we performed another experiment using DQ Red BSA (DQ-BSA, an internally quenched bodipy-labeled Bovine Serum Albumin). This tracer is taken up via endocytosis but only emits fluorescence when undergoing limited proteolysis, thus labeling actively functioning endolysosomes [42]. In controls, DQ-BSA is transported to the endolysosomes after a 5 min pulse followed by 30 min chase, where it displays a punctate pattern (Figure 1I). In contrast, only a few puncta were detectable in Sec20 RNAi cells (Figure 1J and Appendix A), indicating impairment of lysosomal function.

Since the phenotype of Sec20 knockdown resembles that of HOPS tethering complex mutants, we aimed to determine whether HOPS recruitment to late endosomes is impaired by Sec20 depletion. For this purpose, we used Upstream activating sequence (UAS)-driven Vps41/Lt (Vacuolar Protein Sorting41/Light) tagged with 9xHA (9x-hemagglutinin). Vps41 is a Rab7 binding subunit of HOPS, which tethers late endsosomes to lysosomes [39] and normally localizes to a subset of late endosomes in control nephrocytes [26]. Interestingly, we found that Vps41-9xHA sustains its late endosomal localization in cells undergoing Sec20 RNAi. Since we have previously shown that Vps41-9xHA only localizes to late endosomes when HOPS is fully assembled [26], these results indicate that HOPS is functional in Sec20 KD cells.

Given the HOPS- and Rab2-like phenotypes regarding endocytosis as these are also essential factors during autolysosome formation [25,41], we next tested the role of Sec20 in autophagy regulation as well. Our first goal was to obtain a qualitative image of autophagic compartments in Sec20-depleted fat cells. We used mosaic animals where GFP-positive cells were undergoing Sec20 RNAi in combination with the fluorescent reporter 3x-mCherry-Atg8a (expressed in all cells) that marks all autophagic structures including autophagosomes and autolysosomes. We found that small, bright 3x-mCherry-Atg8a dots accumulate in GFP+ Sec20 RNAi cells compared to surrounding control cells (again using multiple RNAi constructs), suggesting an autophagy defect (Figure 2A,B,E,F). We also looked at the level of endogenous Atg8a by immunostaining, which mainly marks phagophores and autophagosomes as Atg8a is degraded in autolysosomes [43]. Atg8a-positive vesicles accumulated in GFP+ Sec20 RNAi cells (Figure 2C), indicating the excess of autophagosomes or non-degrading autolysosomes in these cells. A degradation or transport problem is also supported by the observation that LysoTracker Red-positive, hence acidic lysosomes are much fainter and fewer in GFP+ Sec20 silenced cells compared to their neighboring control cells (Figure 2D). In line with this, Sec20 RNAi cells retain GFP fluorescence of the mCherry-GFP-Atg8a tandem autophagy flux reporter, unlike control cells in which GFP is quenched in lysosomes (Figure 2G,H; note that only GFP fluorescence is lysosomal acid-sensitive, mCherry fluorescence is not [43]). Moreover, the increased levels of the autophagic scaffold protein p62 that is selectively degraded by autophagy [35,43,44] also indicate a failure of autophagic-lysosomal degradation (Figure 2I).

### 3.2. Ultrastructural Analysis Reveals Non-Functional Lysosomes in Fat Cells and Nephrocytes

In the ultrastructure of garland nephrocytes (Figure 3A,B), the most conspicuous finding was the enlargement of late endosomes (also known as α vacuoles, labeled with α), exactly as we anticipated based on our Rab7 immunostaining data. Furthermore, electron-dense endolysomes (labeled by β in control cells) were either small or aberrant in Sec20 depleted nephrocytes. While the small vesicles may represent Golgi-derived vesicles/primary lysosomes failing to be transported to the late endosomes, the large aberrant structures likely correspond to non-functional endolysosomes.

We used mosaic fat tissue of wandering larvae for ultrastructural analysis of fat cells, where control and Sec20 RNAi cells were adjacent and undergoing developmental autophagy (Figure 3C–G). We detected an increased number of autophagosomes and accumulation of small and non-functional autolysosomes that fail to degrade their cargo indicated by the presence of intact ER and mitochondria inside the autolysosomal lumen (enlarged in Figure 3D,E), similar to retromer or V-ATPase loss-of-function cells in which only non-degrading autolysosomes are present [45,46]. The secretory system also shows striking defects, as the ER is enlarged (quantified in Appendix A) with its lumen filled with proteinaceous material compared to the control cells (Figure 3D,E) and the cells also accumulate Golgi-like vesicles (Figure 3F,G) The latter was also confirmed via immunostaining using Golgi-specific GMAP (Golgi microtubule-associated protein) antibody (Appendix A). The ER expansion may result from the failure of ER-resident proteins to recycle to the ER from the Golgi; thus, newly synthesized proteins are trapped in the ER.

### 3.3. Sec20 Promotes Lysosomal Biogenesis and Degradation

As our ultrastructural results implied that autophagosome-lysosome fusion proceeds properly, we aimed to verify it using a fluorescent approach. Thus, we used flies expressing both Lamp1-GFP and 3xmCherry-Atg8a simultaneously as the colocalization of these markers indicates properly formed autolysosomes (Figure 4A). This indeed verified our previous hypothesis, as in Sec20-depleted fat cells, the colocalization of these reporters is evident (Figure 4B), indicating that lysosomal fusions took place normally. Interestingly, Lamp-positive vesicles showed decreased colocalization with the lysosomal hydrolase Cathepsin L (CathL) in Sec20 RNAi cells compared to controls (Figure 4C,D). Furthermore, small Lamp1-GFP-positive autolysosomes also accumulated in Sec20 RNAi cells (Figure 4A,B,E,F). These results imply an important role of Sec20 in proper biosynthetic input to functional lysosomes, considering that autolysosomes can form but they may lack enzymes essential for degradation.

Proteins involved in lysosomal biogenesis have been shown to mediate proper eye pigment granule formation, as pigment granules are lysosome-related organelles [47]. Examples include HOPS tethering complex subunits such as Vps41/Lt, the hypomorphic allele of which causes darker eye color in adult flies [48,49]. When we silenced Sec20 in the developing ommatidia [24], we saw a very similar phenotype in the eyes (Figure 4G,H); thus, we infer that Sec20 also contributes to proper eye pigment granule biogenesis.

### 3.4. Sec20 Acts Independently of Sec22 and Use1 in Lysosomal Regulation but It May Cooperate with Syx18

As Sec20 plays a role in Golgi-ER retrograde transport, we decided to address the question whether the lysosomal degradation defects are due to impaired fusions between COPI vesicles and the ER. We thus silenced the previously described binding partners of Sec20 that are probably involved in Golgi-ER transport. The orthologs of two SNARE proteins, Use1 and Sec22, and tethering protein Dsl1 (Dependence on SLy1-20, known as Zw10 (Zeste white 10) in *Drosophila*) form a functional fusion complex in yeast. Knockdown of any of these proteins neither altered the distribution of 3x-mCherry-Atg8a-positive dots in fat cells (Figure 5A–C,E) nor the size of late endosomes in garland nephrocytes (Figure 6A–D,F). In contrast, RNAi silencing of the third described SNARE partner of Sec20, Syx18, phenocopied the effects of Sec20 knockdown in both garland cells and fat tissue: the size of Rab7-positive vesicles was enlarged in Syx18 loss-of-function nephrocytes (Figure 6E,F) and 3xmCherry-Atg8 accumulated in fat cells (Figure 5D, quantified in Figure 5E).

To confirm the previously described role of Sec20 and its partners in the secretory pathway as well as knockdown efficiencies, we assessed their function in the larval salivary gland. Salivary gland cells produce a large number of secretory (aka. glue) granules in preparation for metamorphosis, the majority of which are secreted at the time of puparium formation to attach the animal to a dry and solid surface away from the wet food where the animals would sink and suffocate during metamorphosis. Residual secretory granules then fuse with lysosomes in a process called crinophagy [30]. We used a reporter system with a glue granule component, Sgs3 (Salivary gland secretion 3), fused with either GFP or DsRed to follow glue granule biogenesis and crinophagy. When glue-containing secretory granules are degraded via crinophagy, GFP is quenched in the acidified lumen of lysosomes while the fluorescence of DsRed is retained. At the beginning of puparium formation in control animals, the majority of glue granules are acidic; thus, they only display red fluorescence, while there are some secretory granules that still show both red and green fluorescence (Appendix A). However, when we silenced Sec20, Syx18, Sec22, Use1 or Zw10 specifically in the salivary gland, mostly diffuse fluorescence was seen in the cytoplasm of the cells for both dsRed and GFP (Appendix A). Furthermore, all of the (pre)pupae fell down from the wall of the culture vials due to the lack of properly formed hence non-secreted glue granules. These observations clearly indicate a failure of proper formation and secretion of glue granules. Thus, all four examined SNARE proteins and Zw10 are responsible for mediating adequate secretion, but only Sec20 and Syx18 appear to be involved in lysosomal biogenesis and regulation.

## 4. Discussion

Lysosomal degradation is a key process during the normal homeostasis of cellular functions, and lysosomal defects potentially lead to severe diseases such as Alzheimer’s disease or various cancers [3,50]. Sec20/BNIP1 is a known mediator of the fusion of COPI-coated, retrograde, Golgi-derived vesicles with the ER; hence, it is responsible for normal ER structure. Moreover, Sec20/BNIP1 is also a positive regulator of apoptosis and a putative positive regulator of mitophagy and mitochondrial fission. Vesicular fusion events are mediated by other factors as well, e.g., tether proteins that anchor the two adjacent membranes, and SNARE proteins (including Sec20) that execute the fusion itself. A tether partner of Sec20 in yeast is Dsl1 (*Drosophila* Zw10) and its SNARE partners include Sec22, Syx18 and Use1.

We showed here that *Drosophila* Sec20 is an important regulator of autophagy and endocytosis via promoting lysosomal degradation and biogenesis. Endocytic and autophagic degradation defects are obvious in garland nephrocytes and in fat cells upon RNAi knockdown of Sec20. Moreover, Sec20 promotes lysosome-related eye pigment granule formation, supporting its role in lysosomal biogenesis in other tissues as well. Similar ER expansion was observed by Sriburi and colleagues, which was caused by unfolded protein response (UPR)-mediated increased ER-biogenesis [51].

We propose that Sec20 regulates autophagy and endocytosis by promoting proper trafficking of lysosomal proteins. Several possibilities exist in which lysosomal proteins can reach lysosomes. For example, it has been shown that Lamp transport depends on Vps41/Lt, while the transport of soluble proteins depends on another HOPS protein Vps18/Dor [52,53]. It has also been demonstrated that Rab2 is responsible for Lamp delivery to endosomes [54], and another study showed that Rab6 is required for Cathepsin targeting to autolysosomes, as similar to Sec20 depleted cell Rab6 mutant cells contain non-degrading autolysosomes [55]. Failure of autophagic degradation caused by impaired autolyosomal enzyme content has been also found upon retromer loss-of-function [45], or in Cathepsin D-deficient mice cells [56]. Accordingly, we propose that Sec20 most likely regulates soluble protein transport to lysosomes, and the Sec20-containing vesicle fusion complex involved in this process is likely different from the Use1-containing one, described in yeast. Based on our salivary gland data, all of the four SNAREs that constitute the Golgi-ER retrograde transport complex in yeast are required for proper secretion, but only two of them, Sec20 and Syx18, are necessary for proper lysosomal degradation in fat cells and nephrocytes. The interaction of Zw10 and Syx18 during the regulation of Golgi-ER trafficking has been characterized in mammalian cells as well [57], suggesting the existence of a conserved Golgi-ER fusion complex. Furthermore, it has been suggested that Sec22 does not regulate autophagy in *Drosophila* as opposed to its conserved role in maintaining normal ER structure [58], which strengthens our findings regarding the dispensable factors described in this work in lysosomal degradation. Thus, a complex consisting of Sec20, Syx18 and perhaps additional so far unidentified SNAREs are likely responsible for the specific trafficking of lysosomal proteins through the ER-Golgi pathway.

## Figures and Tables

**Figure 1 cells-08-00768-f001:**
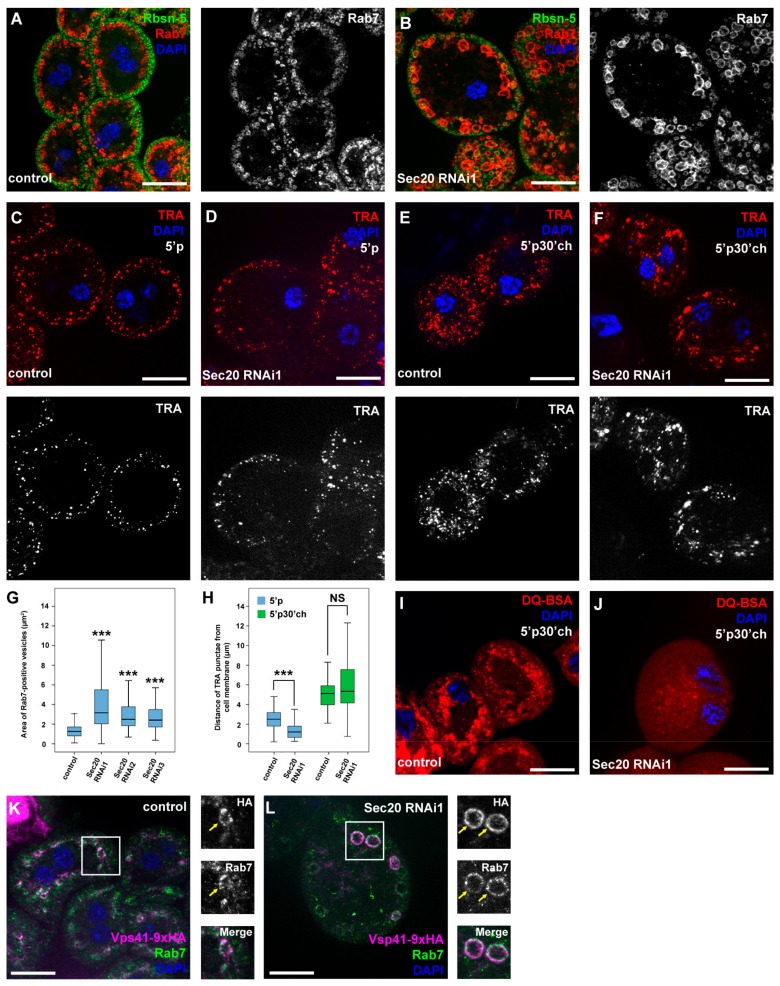
Sec20 is responsible for proper endocytic transport in garland nephrocytes. (**A**, **B**) Garland nephrocytes were co-stained against early endosome-specific Rbsn-5 and late endosome-specific Rab7. Late endosomes show significant enlargement upon Sec20 depletion (**A**) compared to control cells (**B**). (**C**,**D**) When nephrocytes were incubated with TexasRed-Avidin (TRA) fluorescent endocytic tracer for a 5 min pulse (5’p, without subsequent chase), the tracer appeared closer to the cell membrane in Sec20 RNAi cells (**D**) than in controls (**C**). (**E**,**F**) However, when cells were chased for 30 min following the 5 min pulse, TRA was transported to the perinuclear region both in control (**E**) or Sec20 knockdown (**F**) cells. (**G**) Quantification of the areas of Rab7-positive vesicles of the nephrocytes of each Sec20 RNAi lines compared to the same control case (note that the confocal images of Sec 20 RNAi2, RNAi3 RNAi nephrocytes are shown in Appendix A). (**H**) Quantification of the TRA uptake and transport assays from (**C**–**F**). (**G**,**H**). Medians are shown as horizontal black lines within the boxes. Bars show the upper and lower quartiles, and the whiskers plot the smallest and largest observations. (**I**,**J**) Garland nephrocytes were incubated with DQ-BSA during a 5 min pulse and were then chased for another 30 min. In controls, numerous red puncta indicate efficient proteolysis inside endolysosomes (**I**). Significantly fewer puncta were observed in Sec20-depleted cells (**J**). (**K**,**L**) Vps41-9xHA was overexpressed using pros-Gal4 in either control (**K**) or Sec20 RNAi expressing cells (**L**) and stained against Rab7 and HA. The colocalization between Vps41-9xHA with a subset of Rab7 positive endosomes is evident in controls (**K**). Vps41-9xHA sustains its late endosomal localization in cells undergoing Sec20 RNAi. NS: non-significant (*p* ≥ 0.05), ****p* < 0.001. Scale bars: 20 µm.

**Figure 2 cells-08-00768-f002:**
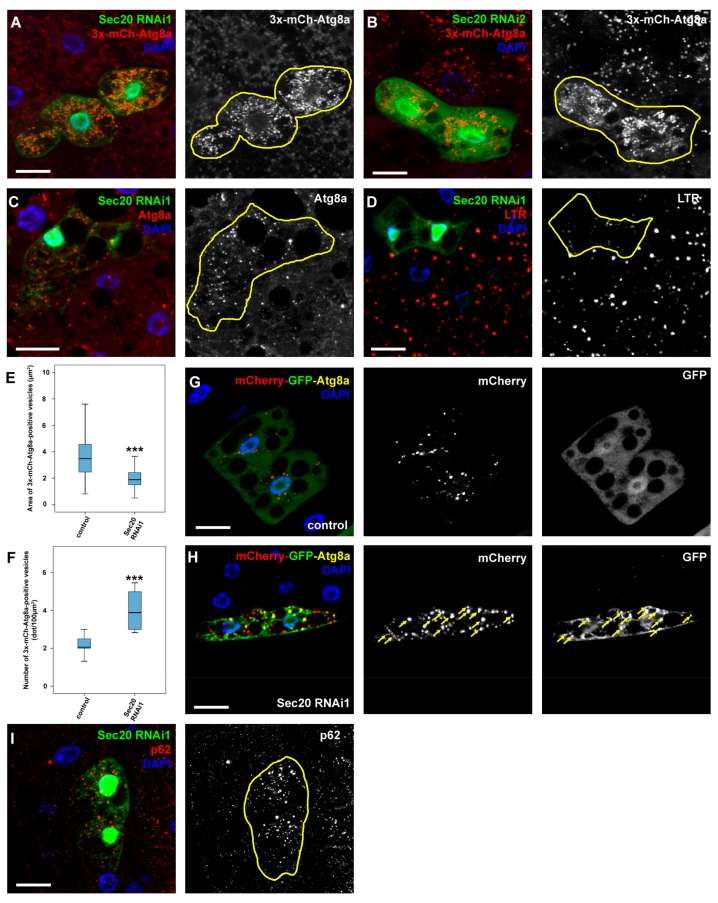
Sec20 is required for autophagic degradation in fat cells of starved larvae. (**A**,**B**) 3xmCherry-Atg8a—a marker that was expressed throughout the fat body and labels both autophagosomes and autolysosomes—revealed an increase in abundancy but decrease in size of autophagic vesicles upon expression of two independent Sec20 RNAi lines in GFP-positive cell clones compared to control cells. (**C**) Endogenous Atg8a (that marks phagophores and autophagosomes) accumulated in Sec20 RNAi cells. (**D**) The acidophilic dye, LTR revealed smaller and less acidic autolysosomes in Sec20-depleted cells. (**E**,**F**) Quantification of the area (**E**) or number (**F**) of the 3xmCherry-Atg8a dots from panel (**A**). Medians are shown as horizontal black lines within the boxes. Bars show the upper and lower quartiles, and the whiskers plot the smallest and largest observations. (**G**,**H**) Tandem mCherry-GFP-Atg8a reporter was expressed either in control (**G**) or Sec20-depleted (**H**) fat cells. While in the former autophagic flux proceeded normally based on the quenching of GFP, in the latter the fluorescence of GFP was retained (arrows) suggesting a failure in autolysosome formation or acidification. (**I**) Endogenous p62 accumulated in GFP+ Sec20 RNAi cells. *** *p* < 0.001, scale bars: 10 µm.

**Figure 3 cells-08-00768-f003:**
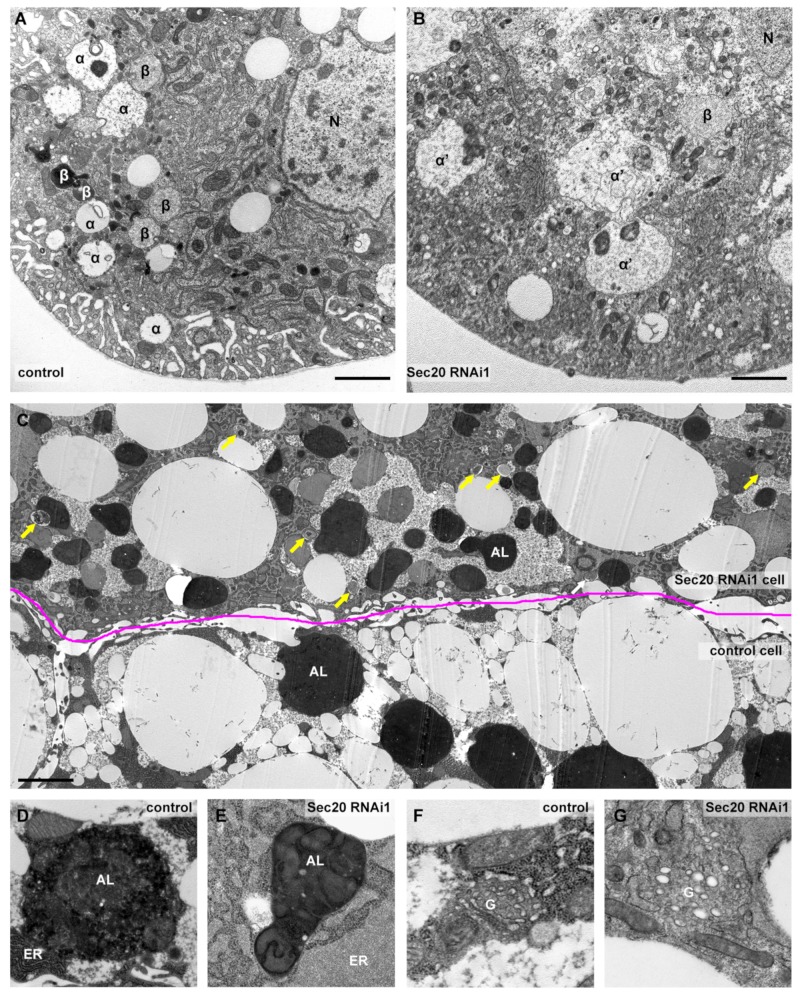
Ultrastructure of Sec20 depleted garland nephrocytes and fat cells. (**A**,**B**) As opposed to control nephrocytes (**A**), late endosomes (marked by α in A or α’ in B) are enlarged upon Sec20 depletion (**B**). (**C**) Compared to a neighboring control fat cell, autophagosomes and autolysosomes accumulate in a Sec 20 RNAi cell of a wandering animal. (**D**,**E**) Enlarged images show representative autolysosomes of control (**D**) and the Sec20-depleted cell (**E**), Compared to control (**D**,**F**), the ER is swollen (**E**) and Golgi is fragmented (**G**) in the knockdown cell. α: late endosome, α’: aberrant late endosome, β: endolysosome, arrows: autophagosomes, AL: autolysosome, ER: endoplasmic reticle, G: Golgi apparatus. Scales: 1 µm in panels (**A**–**C**), 0.5 µm in panels (**D**–**G**).

**Figure 4 cells-08-00768-f004:**
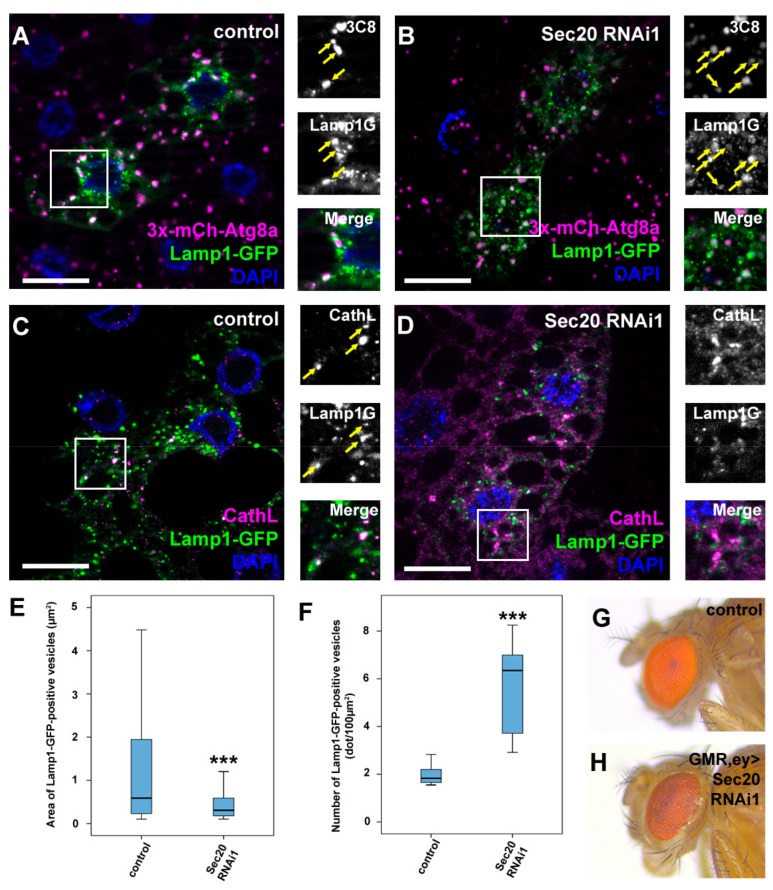
Sec20 is essential for proper lysosomal biogenesis but not for autophagosome-lysosome fusion. (**A**,**B**) Starved fat cells expressing Lamp1-GFP and 3xmCherry-Atg8a. Colocalizing dots (arrows) are evident both in control (A) or Sec20 RNAi (B) cells, suggesting autophagosome-lysosome fusion proceeds normally in both cases. (**A**). (**C**,**D**) Lamp1-GFP-expressing fat cells were immunostained against CathL to detect lysosomes. In control cells (**C**), Lamp1-GFP clearly colocalizes with CathL (arrows), while their overlap is decreased upon Sec20 knockdown (**D**) suggesting a failure in lysosome biogenesis. (**E**,**F**) Quantification of the area (**E**) and number (**F**) of Lamp1-GFP dots showing a significant decrease in size but increase in number of autolysosomes. Medians are shown as horizontal black lines within the boxes. Bars show the upper and lower quartiles, and the whiskers plot the smallest and largest observations. (**G**,**H**) The red color of the compound eye of a control fly (**G**) becomes darker upon eye-specific expression of Sec20 RNAi (**H**) suggesting disturbed biogenesis of lysosome-related eye pigment granules. Combined GMR- and ey-Gal4-expressing flies in wild type background were used as controls (**G**) *** *p* < 0.001, scale bars: 10 μm.

**Figure 5 cells-08-00768-f005:**
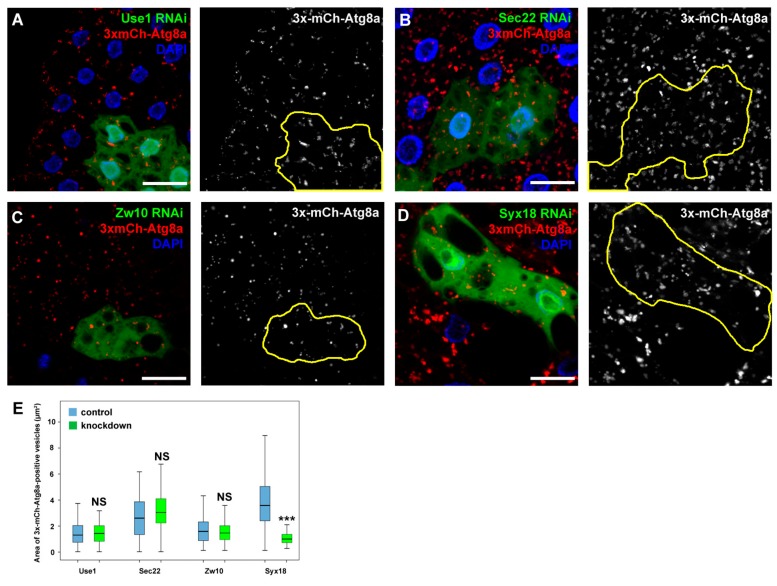
Syx18 may function together with Sec20 in autophagy regulation. (**A**–**E**) Use1 (**A**), Sec22 (**B**), Zw10 (**C**), or Syx18 (**D**) was silenced in the GFP+ cells, and autophagy was assessed by 3xmCherry-Atg8a expressed ubiquitously. Solely the knockdown of Syx18 altered the distribution of the reporter, raising the possibility that it is the only one of the previously known partners of Sec20 that also regulates autophagy. (**E**) Quantification of the area of 3xmCherry-Atg8a dots. Medians are shown as horizontal black lines within the boxes. Bars show the upper and lower quartiles, and the whiskers plot the smallest and largest observations. NS: non-significant (*p* ≥ 0.05), *** *p* < 0.001, scale bars: 10 μm.

**Figure 6 cells-08-00768-f006:**
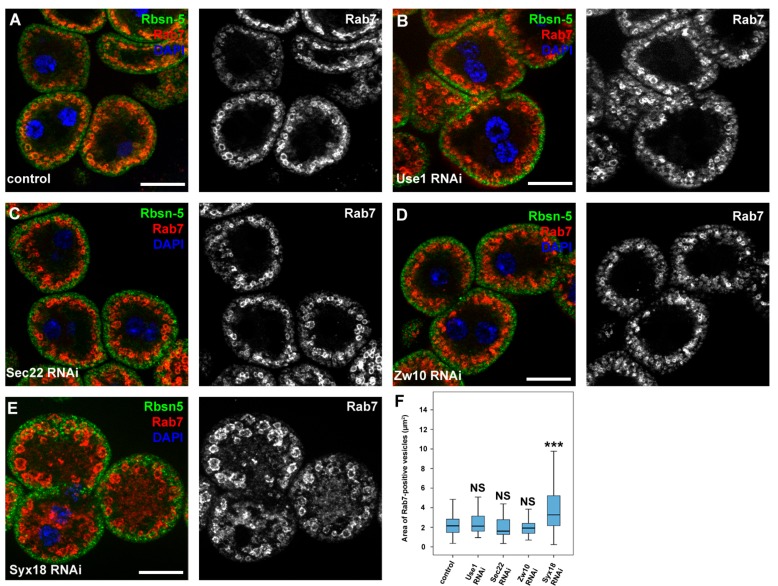
Syx18 regulates endocytic trafficking and degradation. (**A**–**E**) Control (**A**), Use1- (**B**), Sec22- (**C**), Zw10- (**D**) and Syx18-depleted (**E**) garland nephrocytes were immunostained against early endosomal Rbsn-5 and late endosomal Rab7. Only Syx18 depletion caused enlargement of late endosomes similar to Sec20 RNAi nephrocytes (see Figure 1A,B). (**F**) Quantification of the areas of Rab7-positive vesicles. Medians are shown as horizontal black lines within the boxes. Bars show the upper and lower quartiles, and the whiskers plot the smallest and largest observations. NS: non-significant (*p* ≥ 0.05), *** *p* < 0.001, scale bars: 20 μm.

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
