# Peer review of "Sec20 Is Required for Autophagic and Endocytic Degradation Independent of Golgi-ER Retrograde Transport"

_cells, 2019, doi:10.3390/cells8080768_

Round 1
Reviewer 1 Report
In this work, the authors describe a role of Sec20/bNIP1 protein, a SNARE protein involved mainly in vesicular ER-Golgi traffic, in endosome-lysosome and autophagosome-lysosome fusion, thus regulating endosomes degradation and autophagy. In addition, they claim that this role of sec20 is independent of Golgi-ER retrograde transport and it likely acts controlling lysosomal activity. The entire work is done in drosophila models. Some considerations:
The work is interesting and developed in an elegant way. The morphological results are very clear and the figures are nice. However, in my opinion, the main lack to sustain the conclusions is to show a measure of the lysosome activity with and without sec20. If they were right, with this result the work would be quite closed.
They used larvae with Sec depletion in which they find enlarged late-endosomes (and also swelled cells). They interpreted that endosomes are deficiently degraded by lysosomes. Could be another explanation? If endosomes and cells are both enlarged, could not be due to an edema process by water imbalance?
They used starved larvae to study the autophagy and they described an increased number but a decreased size of LC3/Atg8 vesicles. Together with an accumulation of p62 in sec lacking cells, they interpreted that autophagy may be deficient. They show results by immunofluorescence and electron microscopy. However, by themselves, these results are not sufficient to say that autophagy does not work. The autolysosome shown in Fig 3E seems more healthy than the control one (3D).
They show later a lack of colocalization between lamp-positive cells and cathepsine-L and they deduce that autolysosomes can form but they may lack enzyme essential for degradation. That needs some checking.
The results shown are basically morphologic ones and the authors claim several times that they resemble the results described with HOPS loss-of-function. It would have been easy to check in their model the expression of HOPS by WB.
The explanation is sometimes a bit cryptic and even telegraphic, so that it is not always easy to follow.
In conclusion, all these considerations could be summarized as the work has sufficient quality but it lacks a final proof indicating a lack of function of lisosomes.
Author Response
Response to Reviewer 1
We thank the expert Reviewer for the useful comments and suggestions. We have addressed each of the comments and carried out new experiments and included new data when requested. Accordingly, changes have been made to the text and figures, including the correction of minor errors and typos, and the Materials and Methods part was also updated. Please see below for details.
In this work, the authors describe a role of Sec20/bNIP1 protein, a SNARE protein involved mainly in vesicular ER-Golgi traffic, in endosome-lysosome and autophagosome-lysosome fusion, thus regulating endosomes degradation and autophagy. In addition, they claim that this role of sec20 is independent of Golgi-ER retrograde transport and it likely acts controlling lysosomal activity. The entire work is done in drosophila models. Some considerations:
Thank you for the positive initial evaluation of our work.
The work is interesting and developed in an elegant way. The morphological results are very clear and the figures are nice. However, in my opinion, the main lack to sustain the conclusions is to show a measure of the lysosome activity with and without sec20. If they were right, with this result the work would be quite closed.
Thank you for this insight, we performed an uptake assay using DQ-BSA. This is an internally quenched fluorescent tracer and on hydrolysis to single, dye-labeled peptides by proteases, this quenching is relieved. Therefore t the tracer only emits fluorescence in functional endolysosomes. We found that in Sec20 depleted cells the number of fluorescent puncta decreases and this result is now included in Figure 1 (page 6) , and the text has also been changed accordingly. Please see lines 186-192 (page 7) for details.
They used larvae with Sec depletion in which they find enlarged late-endosomes (and also swelled cells). They interpreted that endosomes are deficiently degraded by lysosomes. Could be another explanation? If endosomes and cells are both enlarged, could not be due to an edema process by water imbalance?
Thank you for this suggestion, however the ultrastructure of mitochondria and nuclei seem to be normal we would exclude water imbalance as an explanation which could be caused by osmotic stress or even improper fixation. Please note that we and also others observed that upon inhibited endosomal traffic cells usually swell and become filled with abnormal endocytic/lysosomal vesicles. Accordingly we added the following sentence to the text: These phenotypes point to the impairment of endolysosomal degradation as nephrocytes carrying mutations affecting endosomal traffic often swell due to the continuous uptake of extracellular fluid which they fail to degrade (Page 4, lines 152-155).
Please see Kim et al 2010, PMID: 20194640, Lőrincz et al 2016 PMID: 27253064, Lund et al 2018 PMID: 29940804, Lőrincz et al 2019 PMID: 31194677 for examples and details.
They used starved larvae to study the autophagy and they described an increased number but a decreased size of LC3/Atg8 vesicles. Together with an accumulation of p62 in sec lacking cells, they interpreted that autophagy may be deficient. They show results by immunofluorescence and electron microscopy. However, by themselves, these results are not sufficient to say that autophagy does not work. The autolysosome shown in Fig 3E seems more healthy than the control one (3D).
Thank you for this idea. As tandem mCherry-GFP-Atg8a is an accepted and widely used reporter to examine autophagy progression (please see Lőrincz et al 2017 PMID: 28704946), the impaired autophagic flux (Figure 2G, H) observed in Sec20 depleted cells together with elevated p62 levels both point to degradation failure. This was confirmed by electron microscopy: in fat cells, organelles targeted for degradation swiftly disintegrate and lose their features inside autlysosomes which makes their identification challenging. Thus intact and identifiable organelles seen inside the autolysosomes of Sec20 deficient cells represent lysosome impairment. Please note that very similar autolysosomes were previously seen in v-ATPase- or retromer-depleted cells due to impaired degradation, see Mauvezin et al 2015 PMID: 25959678 and Maruzs et al 2015 PMID: 26172538.
They show later a lack of colocalization between lamp-positive cells and cathepsine-L and they deduce that autolysosomes can form but they may lack enzyme essential for degradation. That needs some checking.
Thank you for the suggestion. The impairment of autophagy due to the lack of Cathepsin have been already shown by Maruzs et al 2015 PMID: 26172538 and Ayala et al 2018 PMID: 30111579. Moreover, as was mentioned above using DQ-BSA, we suggest that endolysosomes in nephrocytes lack the proper amount of enzymes. Therefore, we conclude that similar to endolysosomes, autolysosomes also lack enzymes which is supported by the fluorescent and electronmicroscopic loss of function data.
The results shown are basically morphologic ones and the authors claim several times that they resemble the results described with HOPS loss-of-function. It would have been easy to check in their model the expression of HOPS by WB.
Thank you for suggesting to examine HOPS complex in Sec20 depleted animals. As we lack available HOPS specific antibody (please note that Vps18, Vps16A, Vps33A are also subunits of another complex, named miniCORVET), we used Vps41-9xHA expressing nephrocytes. Normally this reporter localizes to mature late endosomes only when the HOPS holocomplex is fully assembled (Please see Lőrincz et al 2019 PMID: 31194677). Therefore, Vps41 retaining its late endosomal localization in Sec20-depleted cells infers a normal expression pattern of each HOPS subunits and indicates that the recruitment of HOPS proceeds normally in these cells. Please see Figure 2 K,L and page 5, lines 196-203 for details.
Additionally, as the expression of the reporter could not rescue the phenotype of Sec20 depletion (Vps41, Rab7 double positive endosomes were larger than controls), we think that the phenotype of Sec20 knock-down is independent of HOPS.
Reviewer 2 Report
MAJOR COMMENTS
The findings presented in this paper are of good quality, interest and generally appear convincing. However, it is also a "routine" piece of work and rather rigidly adheres to a canonical set of experimental strategies followed many times earlier by this group, limited to the "universe" of autophagy. I would have wished for a stronger effort to pursue the mechanism underlying the function of Sec20.
In my opinion the paper is not acceptable in its present form. I have the following remarks.
It is stated (line 248) that "These results imply an important role of Sec20 in proper biosynthetic input to functional lysosomes considering that autolysosomes can form but they may lack enzymes essential for degradation". However, what ideas the authors have concerning the exact role of Sec20 appear very vague. A more systematic dicussion of the possible involvement of the various routes from Golgi to the endo-lysosomal pathway is requested, based on the framework of, e.g., these papers ([1-3].
From an experimental viewpoint, one information that is completely missing is localization data concerning Sec20 in flies. Was it, e.g., attempted to generate immunohistochemical data or an epitope-tagged UAS/genomic transgenic construct of Sec20?
The authors infer from yeast data that Use1 and Sec22, and tethering protein Dsl1/Zw10, are involved in Golgi-ER transport. However, they do not demonstrate that this is also the case in the fly.
There is no quantification of the EM data, e.g., of the fragmented Golgi or the swollen ER upon Sec20 KD (indeed, the fragmentation of the Golgi is far from obvious). At the very least, several more examples (micrographs) of these effects should be shown.
p8, line 235: "This deterioration of the secretion apparatus looks exactly as expected upon the impairment of a protein responsible for Golgi-ER retrograde trafficking." As with several other claims, this should be substantiated with more references/data involving positive controls with other proteins known to have this effect.
Of some concern is the lack of rescue experiments (e.g., using RNAi-insensitive constructs) to verify that the effects of knocking down Sec20 are specific, in particular because the reported findings are not corroborated using deletion mutants. The use of alternative RNAi constructs reduces but does not eliminate the risk of off-target effects. For at least some of the main results, e.g., those in Figure 1, a rescue would be desired.
In general, more references should be added. For example, the strong claim is made that "lysosomal defects potentially lead to severe diseases such as Alzheimer’s disease or various cancers" (p. 11, l. 314) without any references.
MINOR COMMENTS
In Fig. 1, there is no quantification to back up the statement that early endosomes are not enlarged after Sec20 KD. The Rbsn-5 and/or 5'p TRA-positive endosomes should also be quantified.
p2, line 49: "Sec20 may also have a role in mitophagy as it localized partly to mitochondria and its overexpression resulted in the fragmentation of the mitochondrial network [19,20]."
The authors may consider changing this sentence from past to the present tense as in the surrounding text.
p2, line 76: The section beginning "In order to acquire starving fat cells…" might deserve a separate headline (is not part of the fly strain description).
p3, line 114: The following sentence is unclear and grammatically flawed, and should be rephrased: "In spite of increasing images from 8 consecutive focal planes (section 114 thickness: 0.25 μm for nephrocytes and 0.35 μm for fat cells) were projected onto one single image, 115 except for the colocalization assays, where we aimed to exclude any false positive colocalization."
p3, line 137: "10-10 cells 137 of each genotype were quantified" presumably should be 10-100?
p4, line 141:" The normality test for each data sets were performed using SPSS17 (IBM), and as the distribution of at least one of the compared datasets were non-Gaussian in each analysis, we either performed the Mann-Whitney U-test 143 for two samples or the Kruskal-Wallis test with post-hoc pairwise analysis for more than two samples."
This is fine. However, as a general advice, I wish to point to author's attention to an alternative when the data are skewed to the right (as in, e.g., Fig 4E): the use of a logarithmic or square root transformation before performing t-tests or ANOVA. This procedure may sometimes yield better statistical power than non-parametric methods.
p5, line 169: "Bars show the upper and lower quartiles, and the whiskers indicate SEM". Upper and lower whiskers can hardly indicate SEM as they differ in length. Do they indicate max and min values? Please explore and correct.
p5, line 171: "This phenotype most likely results from the continuous input to late endosomes in the absence of proper lysosomal fusion or degradation". It is suggested to add some references to back up this claim.
Figure 1, e.g. panel D and F suffer heavily from conversion to PDF in terms of pixelation and distortion. Perhaps this is only a problem in my reviewers' copy but should be checked in the proof.
It is suggested to include a summary figure to provide an overview, e.g., of the differential roles of Sec20, Syx18, Sec22, Use1 or Zw10 in secretion and lysosomal biogenesis and regulation.
What is the control genotype ,e.g., in Fig. 4G? This should be stated in the figures or in a supplementary list of genotypes if that has not been produced already.
1. Lund, V.K., et al. Autophagy, 2018. 14(9): p. 1520-1542.
2. Sriram, V., et al. J Cell Biol, 2003. 161(3): p. 593-607.
3. Swetha, M.G., et al. Traffic, 2011. 12(8): p. 1037-55.
Author Response
Response to Reviewer 2
We thank the expert Reviewer for the useful comments and suggestions. We have addressed each of the comments Accordingly, changes have been made to the text and figures, including the correction of minor errors and typos, and the Materials and Methods part was also updated. Please see below for details.
The findings presented in this paper are of good quality, interest and generally appear convincing. However, it is also a "routine" piece of work and rather rigidly adheres to a canonical set of experimental strategies followed many times earlier by this group, limited to the "universe" of autophagy. I would have wished for a stronger effort to pursue the mechanism underlying the function of Sec20.
In my opinion the paper is not acceptable in its present form. I have the following remarks.
It is stated (line 248) that "These results imply an important role of Sec20 in proper biosynthetic input to functional lysosomes considering that autolysosomes can form but they may lack enzymes essential for degradation". However, what ideas the authors have concerning the exact role of Sec20 appear very vague. A more systematic dicussion of the possible involvement of the various routes from Golgi to the endo-lysosomal pathway is requested, based on the framework of, e.g., these papers ([1-3].
Thank you for your valuable insights. We added a more thorough discussion about the potential role of Sec20 based on the articles you mentioned and several others, please see pages 12-13, lines 357-377 for details.
From an experimental viewpoint, one information that is completely missing is localization data concerning Sec20 in flies. Was it, e.g., attempted to generate immunohistochemical data or an epitope-tagged UAS/genomic transgenic construct of Sec20?
Thank you for suggesting to generate immunohistochemical data or an epitope-tagged UAS/genomic transgenic construct of Sec20. Both UAS and genomic promoter-driven GFP-Sec20 construct were attempted to create, but unfortunately the transgenes were not functional. Unfortunately, the given timeframe of revision is not enough for a second attempt. Also, an antibody against Drosophila Sec20 is not available, and due to sequence differences the antibodies used for mammalian BNIP1 are probably not applicable.
The authors infer from yeast data that Use1 and Sec22, and tethering protein Dsl1/Zw10, are involved in Golgi-ER transport. However, they do not demonstrate that this is also the case in the fly.
The Drosophila larval salivary gland is a widely used model studying Glue-protein (Sgs) secretion (see reviews of Biyasheva et al. 2001, PMID: 11180965, Farkaš 2015, PMID: 25960390 and Tran and Ten Hagen 2017, PMID: 28302911). As the salivary glands of Use1, Sec22 and Dsl1/Zw10 depleted animals devoid of Sgs3 granules we conclude that the secretion apparatus is most likely damaged.
There is no quantification of the EM data, e.g., of the fragmented Golgi or the swollen ER upon Sec20 KD (indeed, the fragmentation of the Golgi is far from obvious). At the very least, several more examples (micrographs) of these effects should be shown.
Thank you for suggesting to quantify our data, accordingly we measured the diameter of ER cisterns in control and Sec20 KD cells and show the data in Figure S3. Furthermore, we no longer state that the Golgi is fragmented and now we state “the cells also accumulate Golgi-like vesicles”.
p8, line 235: "This deterioration of the secretion apparatus looks exactly as expected upon the impairment of a protein responsible for Golgi-ER retrograde trafficking." As with several other claims, this should be substantiated with more references/data involving positive controls with other proteins known to have this effect.
Thank you for this suggestion, to avoid confusion we removed that sentence from the text.
Of some concern is the lack of rescue experiments (e.g., using RNAi-insensitive constructs) to verify that the effects of knocking down Sec20 are specific, in particular because the reported findings are not corroborated using deletion mutants. The use of alternative RNAi constructs reduces but does not eliminate the risk of off-target effects. For at least some of the main results, e.g., those in Figure 1, a rescue would be desired.
Thank you for suggesting to perform rescue experiments. However, as our recombinant Sec20-GFP construct was not functional, we were not able to perform rescue experiments. Nonetheless, we used 3 independent RNAi constructs from 3 different stock centres and we find it unlikely that the observed three identical phenotypes are due to off-target effects. Unfortunately, the given timeframe of revision is also not enough for rescue experiments.
In general, more references should be added. For example, the strong claim is made that "lysosomal defects potentially lead to severe diseases such as Alzheimer’s disease or various cancers" (p. 11, l. 314) without any references.
Thank you for this suggestion, accordingly new references were added to the text (page 12, line 344).
MINOR COMMENTS
In Fig. 1, there is no quantification to back up the statement that early endosomes are not enlarged after Sec20 KD. The Rbsn-5 and/or 5'p TRA-positive endosomes should also be quantified.
Thank you for this suggestion. Early endosome size has been quantified and this data is shown in Figure S2.
p2, line 49: "Sec20 may also have a role in mitophagy as it localized partly to mitochondria and its overexpression resulted in the fragmentation of the mitochondrial network [19,20]."
The authors may consider changing this sentence from past to the present tense as in the surrounding text.
Thank you, this sentence is changed to present tense (can be now found on page 2, line 49).
p2, line 76: The section beginning "In order to acquire starving fat cells…" might deserve a separate headline (is not part of the fly strain description).
Thank you, this section has a separate headline in the revised manuscript.
p3, line 114: The following sentence is unclear and grammatically flawed, and should be rephrased: "In spite of increasing images from 8 consecutive focal planes (section 114 thickness: 0.25 μm for nephrocytes and 0.35 μm for fat cells) were projected onto one single image, 115 except for the colocalization assays, where we aimed to exclude any false positive colocalization."
Thank you for finding this error, we accidentally left “In spite of increasing” in that sentence, which now is removed from the text.
p3, line 137: "10-10 cells 137 of each genotype were quantified" presumably should be 10-100?
We originally intended to state that in the case of every genotypes we always measured 10 cells. Therefore 10-10 did not indicate an interval. In order to avoid confusion, the original part of that sentence “10-10 cells of each genotype were quantified” was changed to “10 cells of each genotype were quantified”.
p4, line 141:" The normality test for each data sets were performed using SPSS17 (IBM), and as the distribution of at least one of the compared datasets were non-Gaussian in each analysis, we either performed the Mann-Whitney U-test 143 for two samples or the Kruskal-Wallis test with post-hoc pairwise analysis for more than two samples." This is fine. However, as a general advice, I wish to point to author's attention to an alternative when the data are skewed to the right (as in, e.g., Fig 4E): the use of a logarithmic or square root transformation before performing t-tests or ANOVA. This procedure may sometimes yield better statistical power than non-parametric methods.
Thank you for this insight, we will keep that suggestion in mind in the future.
p5, line 169: "Bars show the upper and lower quartiles, and the whiskers indicate SEM". Upper and lower whiskers can hardly indicate SEM as they differ in length. Do they indicate max and min values? Please explore and correct.
Thank you for finding that error. Indeed they indicate maximum and minimum values. Accordingly all figure legends have been corrected and we state: “the whiskers plot the smallest and largest observations”.
p5, line 171: "This phenotype most likely results from the continuous input to late endosomes in the absence of proper lysosomal fusion or degradation". It is suggested to add some references to back up this claim.
Thank you, we added the sentence to the text: “These phenotypes point to the impairment of endolysosomal degradation as nephrocytes carrying mutations affecting endosomal traffic often swell due to the continuous uptake of extracellular fluid which they fail to degrade” (Page 4, lines 152-155) with appropriate references.
It is suggested to include a summary figure to provide an overview, e.g., of the differential roles of Sec20, Syx18, Sec22, Use1 or Zw10 in secretion and lysosomal biogenesis and regulation.
Thank you for suggesting to add a summary figure to the text, however, in the absence of localization data we would like to avoid preparing a summary figure, which might be too speculative.
What is the control genotype ,e.g., in Fig. 4G? This should be stated in the figures or in a supplementary list of genotypes if that has not been produced already.
Thank you for suggesting to include the genotype of control, accordingly we added this to the figure legend.
Round 2
Reviewer 1 Report
In my opinion, the authors have done efforts enough to ameliorate the work and it is now suitable for publication in Cells.